# EMP2 Serves as a Functional Biomarker for Chemotherapy-Resistant Triple-Negative Breast Cancer

**DOI:** 10.3390/cancers16081481

**Published:** 2024-04-12

**Authors:** Ann M. Chan, Brian Aguirre, Lucia Liu, Vei Mah, Justin M. Balko, Jessica Tsui, Navin P. Wadehra, Neda A. Moatamed, Mahdi Khoshchehreh, Christen M. Dillard, Meagan Kiyohara, Yahya Elshimali, Helena R. Chang, Diana Marquez-Garban, Nalo Hamilton, Richard J. Pietras, Lynn K. Gordon, Madhuri Wadehra

**Affiliations:** 1Department of Pathology Lab Medicine, David Geffen School of Medicine, University of California, Los Angeles, CA 90095, USAvmah@ucla.edu (V.M.);; 2UCLA Stein Eye Institute and the Department of Ophthalmology, David Geffen School of Medicine, UCLA, Los Angeles, CA 90095, USA; 3Department of Medicine, Vanderbilt University Medical Center, Nashville, TN 37232, USA; 4Division of Cancer Research and Training, Department of Internal Medicine, Charles Drew University of Medicine and Science, 1720 East 120th Street, Los Angeles, CA 90059, USA; 5Division of Surgical Oncology, Department of Surgery, David Geffen School of Medicine, UCLA, Los Angeles, CA 90095, USA; 6Jonsson Comprehensive Cancer Center, David Geffen School of Medicine, UCLA, Los Angeles, CA 90095, USA; 7Division of Hematology and Oncology, Department of Medicine, David Geffen School of Medicine, UCLA, Los Angeles, CA 90095, USA; 8School of Nursing, UCLA, Los Angeles, CA 90095, USA

**Keywords:** triple-negative breast cancer, chemotherapy, epithelial membrane protein 2, biomarker

## Abstract

**Simple Summary:**

Breast cancer (BC) ranks as among the top two commonly diagnosed cancers in women worldwide. Triple-negative breast cancer (TNBC) is recognized globally as the most lethal subtype of BC, with patients often building resistance to conventional chemotherapy treatments. This study assesses the relationship between taxane-based chemotherapy resistance in BC and the oncoprotein epithelial membrane protein 2 (EMP2), with an emphasis on the aggressive TNBC group. Analyses of preclinical models and human tissue samples reveal that EMP2 may not only be used to predict patient response to taxanes but also serve as a therapeutic target for resistant disease. Our findings demonstrate a correlation between elevated EMP2 expression post-chemotherapy and reduced survival outcomes and reveal that targeting EMP2 in conjunction with chemotherapy yields a synergistic reduction in TNBC tumor volume. Thus, results shed light on a novel biomarker for guiding the personalized, targeted treatment of intractable and recurrent TNBC.

**Abstract:**

Breast cancer (BC) remains among the most commonly diagnosed cancers in women worldwide. Triple-negative BC (TNBC) is a subset of BC characterized by aggressive behavior, a high risk of distant recurrence, and poor overall survival rates. Chemotherapy is the backbone for treatment in patients with TNBC, but outcomes remain poor compared to other BC subtypes, in part due to the lack of recognized functional targets. In this study, the expression of the tetraspan protein epithelial membrane protein 2 (EMP2) was explored as a predictor of TNBC response to standard chemotherapy. We demonstrate that EMP2 functions as a prognostic biomarker for patients treated with taxane-based chemotherapy, with high expression at both transcriptomic and protein levels following treatment correlating with poor overall survival. Moreover, we show that targeting EMP2 in combination with docetaxel reduces tumor load in syngeneic and xenograft models of TNBC. These results provide support for the prognostic and therapeutic potential of this tetraspan protein, suggesting that anti-EMP2 therapy may be beneficial for the treatment of select chemotherapy-resistant TNBC tumors.

## 1. Introduction

Breast cancer (BC) remains among the top two cancers diagnosed in women worldwide. Despite the significant improvement in screening and management, the incidence of BC has continued to rise over the past two decades [1]. Regarded as a clinically and genetically heterogeneous disease, BC is classified based on the positive or negative expression of estrogen receptor (ER), progesterone receptor (PR) and human epidermal growth factor (HER2). One aggressive subtype that disproportionately impacts racial and ethnic minority groups, particularly Black and Hispanic premenopausal women [2], is triple-negative BC (TNBC). Defined by the absence of ER, PR, and HER2 expression, TNBC accounts for 15–20% of all diagnosed BCs, but over the last decade, a steady increase has been reported in its incidence [3].

While the availability of new treatment options has improved the mortality associated with BC, many protocols utilize taxanes, especially paclitaxel and docetaxel, as the chemotherapeutic backbone used to treat the disease [4]. Current guidelines have established taxanes as an important therapeutic option in both early and metastatic BC in multiple subtypes, including TNBC and those with HER2+ disease [4,5,6]. TNBC treatment predominantly involves cytotoxic chemotherapy, with taxanes and anthracyclines as the standard of care in both the neoadjuvant and adjuvant settings [7,8]. However, even in early stage TNBC, only 30–40% of patients treated with standard neoadjuvant chemotherapy achieve a pathologic complete response (pCR) post-treatment [9]. Some patients even show disease progression while receiving taxane treatment [10,11]. Moreover, responses to these treatments are often short-lived, with patients exhibiting a median overall survival of 12 to 18 months [12].

The frequent occurrence of taxane resistance poses a major hurdle in managing TNBC and other forms of BC [8]. The precise mechanisms underlying resistance remain elusive, necessitating the exploration of alternative strategies and combination therapies to enhance patient outcomes. Given the diversity of the TNBC patient population, no single therapeutic approach will necessarily be universally effective. Identifying patient stratifications, along with tailored therapies effective for distinct subgroups, will be essential for maximizing treatment success, minimizing time costs and side effects from ineffective treatment, and developing a rational approach to targeted therapy. This challenge highlights the need for identifying prognostic and targetable biomarkers, which will guide clinical decision making and tailored therapeutic approaches [13].

One recently identified biomarker in TNBC is the tetraspan protein epithelial membrane protein 2 (EMP2). EMP2 is expressed in >75% of invasive breast tumors, with over 95% of TNBC patients showing at least some expression of the antigen [14]. EMP2 promotes tumor invasion, appears to regulate tumor-mediated neoangiogenesis, and has been shown to increase the proportion of cancer stem-like cells [14,15]. However, the relationships between EMP2, taxane resistance, and TNBC disease progression have not been investigated.

To investigate the potential of EMP2 as a biomarker in patients with resistant disease, we evaluated its expression in a panel of women with BC pre- and post-standard chemotherapy, with an emphasis on understanding changes in its expression in TNBC. Our findings indicate that high EMP2 levels correlate with poor patient survival outcomes, and we demonstrate a link between taxane treatment and a subsequent increase in EMP2-positive tumor cells. Crucially, our study reveals that anti-EMP2 monoclonal antibodies (mAbs) can act in synergy with taxane-based chemotherapy, reducing tumor load in both syngeneic and xenograft models of TNBC. Thus, EMP2 holds potential as not only a predictive biomarker but also a novel therapeutic target for taxane-resistant TNBC.

## 2. Materials and Methods

### 2.1. Patients and Tumor Specimens

All specimens were routinely processed and diagnosed according to national and international guidelines. Formalin-fixed, paraffin-embedded (FFPE) samples assessed in this study were approved by the Institutional Review Board of the Instituto Nacional de Enfermedades Neoplásicas (INEN 10-018) [16] or UCLA (IRB # 20-001197).

To assess how neoadjuvant chemotherapy (NACT) affects EMP2 expression, initially, forty-two patients with BC were diagnosed and treated at the Instituto Nacional de Enfermedades Neoplásicas in Lima, Peru as previously described [16]. Tumors were characterized based on hormone receptor levels and HER2 overexpression measured by IHC and/or HER2 FISH, and 23 patients were classified as having TNBC. The diagnostic biopsy and the post-NAC surgically resected tumors were curated onto a tissue microarray with linked clinical and pathological data. Samples were scored by two independent breast pathologists masked to clinical information.

To further confirm the relationship between EMP2 expression and TNBC recurrence, an additional 8 patients treated with taxane-based chemotherapy (neoadjuvant or/and adjuvant) between 2013 and 2020 at UCLA were analyzed. Remnant FFPE tumor samples (baseline and residual) were obtained and stained for EMP2. Samples were scored by two independent pathologists (N.M., M.K., or Y.E.) masked to clinical information.

### 2.2. Cell Lines and Cell Culture

4T1 cells are TNBC cell lines derived from Balb/c murine mammary tissue and were obtained from the American Type Culture Collection (ATCC, Manassas, VA, USA). RPMI medium (Thermo Fisher, Waltham, MA, USA) supplemented with 10% fetal calf serum (FCS; Hyclone Laboratories, Logan, UT, USA), 1 mM sodium pyruvate, 2 mM L-glutamine, and 100 U/mL penicillin with 100 U/mL streptomycin (P/S; Life Technologies, Carlsbad, CA, USA) were used to cultivate the cell line, and cells were cultured in a 5% humidified CO_2_ chamber at 37 °C. 4T1 cells were infected with firefly luciferase (FLuc) by the UCLA/JCCC Viral Vector Core Lab. MDA-MB-231 cells are a TNBC cell line obtained from the ATCC. Cells were grown in DMEM medium (Thermo Fisher) supplemented as above. Both cell lines were routinely tested for mycoplasma monthly (Lonza, Bend, OR, USA), and cell lines were routinely evaluated for cellular morphology and growth characteristics. All cells were used within 6 months of resuscitation.

### 2.3. Creation of Taxane Resistant Cells

To create 4T1 taxane-resistant cells for in vitro experiments, 1 × 10^5^ 4T1/FLuc cells were injected into the second mammary fat pad of BALB/c mice (Charles River Laboratories, Wilmington, MA, USA). Tumors were treated with saline or 10 mg/kg docetaxel weekly after reaching 100 mm^3^. When the tumors reached ~100 mm^3^, mice were randomly divided between two groups and treated via intraperitoneal injection (IP) with either saline or pharmaceutical-grade docetaxel (10 mg/kg, weekly; Sandoz, Princeton, NJ, USA). Tumor growth was measured weekly using calipers, with volumes determined using the following formula: (length × width^2^)/2.

To create MDA-MB-231 taxane-resistant cells for in vitro experiments, 4 × 10^6^ wildtype cells were mixed 1:1 in Matrigel Matrix (Corning, Corning, NY, USA) and implanted into the second or third mammary fat pad of BALB/c Nude mice (Charles River Laboratories). When the tumors reached ~200 mm^3^, mice were randomly divided between two groups and treated via intraperitoneal injection (IP) with either saline or docetaxel (20 mg/kg, weekly) as above.

### 2.4. Anti-EMP2 mAbs

Anti-EMP2 mAbs were produced in house [15] or by Lake Pharma (San Francisco, CA, USA) according to their standard practices [14]. A detailed characterization [17] of the mAbs binding and specificity has previously been published [15,18]. The anti-EMP2 mAb detailed here is a fully human IgG1 with a kappa backbone. The antibody batch affinity was determined by Lake Pharma against the human EMP2 peptide using ForteBio Octet and showed an affinity between 5 and 8 nM. Appendix A characterizes a batch of the antibody, showing the reduced forms. The mAb was stored as frozen aliquots in a 200 mM HEPES, 100 mM NaCl, 50 mM NaOAc, pH 7.0 solution prior to thaw.

Additional experimental details can be found in the Appendix A.

## 3. Results

### 3.1. EMP2 Transcript Expression in Chemotherapy Resistant TNBC Tumors

Taxane usage has been widespread in BC for over two decades [4]. Often used as front-line treatment for hormone receptor-positive HER2-negative (HR+/HER2-), HER2-positive, and TNBC disease, taxanes represent one of the most effective chemotherapies used in BC [17]. However, resistance remains a challenge, as it is estimated that one in three patients will eventually develop recurrent or metastatic disease [19]. Thus, there is a crucial need to identify the predictive biomarkers of therapeutic resistance.

As EMP2 has been shown to promote tumor initiation and be a potential marker for stemness [14], we initially queried levels of its transcript using ROC plotter [20] and Kaplan–Meier (KM) plotter [21]. Both web tools compile samples and patient clinical characteristics from The Cancer Genome Atlas (TCGA), Gene Expression Omnibus (GEO), and European Genome-Phenome Archive (EGA) databases, evaluating correlations between gene expression and either therapy response (using ROC plotter) or survival prognosis (using KM plotter). ROC plotter was initially employed to assess EMP2 mRNA expression in 83 node-positive BC patients who either achieved or did not achieve pCR following taxane-based therapy. Tumors from patients who failed to achieve pCR exhibited a significant increase in EMP2 mRNA levels compared to tumors that responded to taxane-based therapy (*p* = 0.03 by Mann–Whitney test; Figure 1A). Given the high recurrence rates associated with TNBC, we next evaluated the expression of EMP2 in this subtype of disease. Twenty TNBC node-positive patients were identified. Tumors from patients who failed to achieve a pCR trended to show an increase in EMP2 mRNA expression compared to tumors that responded to a taxane-based therapy (*p* = 0.09 by Mann–Whitney test; ROC *p* value = 0.004; Figure 1B).

To expand on this observation, the ability of EMP2 levels to predict overall survival or recurrence-free survival (RFS) was evaluated using KM plotter. Initially, EMP2 was evaluated in all subtypes of BC. A total of 275 patients who received any systemic chemotherapy were analyzed, and high expression of EMP2 correlated with poor relapse-free survival (*p* = 0.013; Figure 1C). To further evaluate the utility of EMP2, its expression was evaluated in 116 patients with TNBC (ER-/PR-/HER2-) who received any systemic chemotherapy. Consistent with published reports [14,15], a high expression of EMP2 occurred in ~70% of patients and was correlated with poor overall survival (*p* = 0.021) and poor relapse-free survival (*p* = 0.017; Figure 1D). To further explore this finding, TNBC node-positive patients were analyzed. A high expression of EMP2 was significantly correlated with poor relapse-free survival (*p* = 0.001; Figure 1E).

### 3.2. EMP2 Protein Expression in Chemotherapy Resistant TNBC Tumors

While the results above suggested that EMP2 mRNA may be upregulated in taxane-resistant tumors, no data are currently available on its protein expression in such tumors. To quantify changes in EMP2 after neoadjuvant chemotherapy (NACT), we analyzed 23 matched pre- and post-treatment biopsies on a tissue microarray (TMA) created from the previously described cohort of women with TNBC from Peru [16]. As seen in Appendix A, minimal expression was observed in normal breasts, highlighting the increase in expression normally observed in BC. While no statistical difference was collectively observed between all pre- and post-treatment samples, EMP2 levels increased in 13/23 (56%) patients, while a reduction occurred in 10/23 (43%) patients (Figure 2A and Appendix A). These changes did not correlate with patients’ stage, age, or menopausal status [23] (Table 1). However, the increased expression of EMP2 in post-treatment samples was positively correlated with node-positive disease (*p* = 0.0068; Table 1), indicating a potential involvement of EMP2 in the progression of TNBC.

Treatment for these TNBC patients often consisted of anthracycline or an anthracycline–taxane doublet [16]. When we stratified the data based on the inclusion of a taxane into the treatment protocol, a significant increase in EMP2 expression resulted compared to the tumor pre-treatment group (Figure 2B). Importantly, this increase in expression occurred specifically upon taxane inclusion, as treatment regimens without taxanes did not produce a similar response (Figure 2C). We next analyzed patient response to treatment, and similar to the transcriptomic analysis, patients with an increase in EMP2 post-chemotherapy showed reduced recurrence-free and overall survival compared to tumors with low or negative EMP2 levels following treatment (Figure 2D,E).

Given the small sample size of the TNBC patients who received a taxane as part of their therapeutic protocol, we next evaluated the change in EMP2 post-taxane treatment using an independent cohort of patients treated at UCLA. This group of eight matched TNBC patients received four cycles of Taxotere^®^/carboplatin (75 mg/m^2^/AUC = 6) as neoadjuvant chemotherapy at UCLA, and FFPE samples were procured and stained for EMP2 expression. The mean histological score for EMP2 showed a significant increase in post-taxane treatment samples compared to the naïve tumor (Figure 3A). When patient tumors were matched, an increase in the EMP2 histological score was observed in 6/8 (75%) post neoadjuvant tumors (Figure 3B). Representative images from two patients pre- and post-NACT, compared to staining with an isotype control, are shown (Figure 3C). While the increase in expression observed did not correlate with patients’ stage, age, menopausal status, or ethnicity, the increase in EMP2-positive tumor cells following neoadjuvant therapy significantly correlated with tumor recurrence (*p* = 0.04; Appendix A).

### 3.3. Modeling Chemo-Resistance Using Syngeneic Mouse Models

Our results thus far have demonstrated that high EMP2 mRNA and protein expression correlated with taxane resistance in patients. To determine if taxane treatment alters the number of EMP2-positive cells, we utilized the 4T1 syngeneic mammary model, as other groups have reported that these cells are resistant to taxane-based therapy [24,25]. 4T1/FLuc cells were injected into syngeneic animals and treated with saline or docetaxel weekly. Consistent with previous reports, these cells remained largely refractory to treatment (Figure 4A). We next assessed EMP2 levels between naïve and resistant cells, and similar to the results observed in patient samples, the number of strongly EMP2-positive cells significantly increased post-taxane treatment (Figure 4A, right).

While results suggest that EMP2 may serve as a biomarker for taxane resistance, it is unknown if resistant tumors are responsive to anti-EMP2-based therapy. Our laboratory has created anti-EMP2 mAbs that bind to the second extracellular domain of both murine and human EMP2 [15]. To assess the efficacy of these mAbs, we created two models of taxane resistance using 4T1/FLuc tumor cells (Figure 4). In the first model, we tested the efficacy of concurrent therapy with docetaxel and anti-EMP2 mAbs versus docetaxel and control mAbs once tumors reached >100 mm^3^. In the second model, tumors were treated sequentially. Animals were first treated with docetaxel for two weeks and then anti-EMP2 or control mAb therapy was included.

In the concurrent model, anti-EMP2 mAbs significantly reduced the average tumor volume and post-euthanasia tumor weight compared to those treated with docetaxel and a control mAb (Figure 4B,C). The growth volumes in individual tumors are shown in Appendix A. Next, we evaluated the expression of EMP2 in the tumors post treatment. Within the saline-treated group, modest EMP2 expression within the plasma membrane and cytosol was observed (Appendix A). Treatment with docetaxel and the control IgG significantly increased the number of strongly EMP2-positive tumor cells observed. In contrast, treatment with docetaxel and anti-EMP2 mAbs significantly reduced the relative mean EMP2 expression within the tumor parenchyma (Figure 4C and Appendix A).

Similarly, in the sequential model, two initial treatments with docetaxel followed by the addition of anti-EMP2 mAb significantly reduced the average tumor volume compared to treatment with the control mAbs (Figure 4D). The growth curves for individual mice are shown in Appendix A. To determine if the change in tumor volume correlated with a change in weight, tumors were harvested. A significant reduction in tumor weight was observed in the combination treatment group with the anti-EMP2 mAb compared to the saline and control IgG-treated groups (Figure 4E). To next evaluate EMP2 expression within these tumors, its levels were quantitated using standard immunohistochemistry. Within the docetaxel plus control IgG-treated group, both the number and intensity of EMP2-positive tumor cells appeared to increase (Appendix A). In contrast, treatment with the anti-EMP2 mAbs significantly reduced the number and intensity of EMP2-positive tumor cells, suggesting the specificity of the mAb treatment.

### 3.4. In Vivo Effects Using Human Xenograft Models

Our syngeneic models revealed that the number and/or intensity of EMP2-positive tumor cells increased following taxane treatment. To determine if this effect could be mimicked using human xenograft models, we utilized the triple-negative cell line MDA-MB-231. These BC cells were injected orthotopically into the mammary fat pads of BALB/c Nude female mice, and tumor load was then compared between naïve and taxane-treated tumors. As shown in Figure 5A, MDA-MB-231 tumors were generally insensitive to taxane treatment. We next evaluated the expression of EMP2 in taxane-resistant or naïve tumors. A significant increase in EMP2 intensity and/or the number of EMP2-positive cells was observed post-treatment in these xenograft models (Figure 5A, below).

There is currently a paucity of treatment options for women with therapy-resistant TNBC. In order to determine if anti-EMP2 mAbs may target this reservoir of taxane-resistant cells, MDA-MB-231 tumors treated with docetaxel weekly at 20 mg/kg were created in mice. These largely taxane-resistant tumors were then harvested and disassociated in order to allow for in vitro testing with control or anti-EMP2 mAbs. Using an ATP assay as a measure for cell viability, the treatment of taxane-resistant MDA-MB-231 cells with anti-EMP2 mAbs resulted in a significant increase in cell death compared to treatment with control mAbs (Figure 5B).

Several studies have independently confirmed that the metastatic potential of a tumor is due to a minor subpopulation of cancer cells termed cancer stem cells (CSCs) [26,27]. These cells are defined by their ability to self-renew and are believed to be responsible for driving recurrence and initiating metastasis [26]. To determine if anti-EMP2 mAbs can target refractory stem-like cells, taxane-resistant MDA-MB-231 isolated from above were grown under low adhesion conditions to allow for the formation of “mammospheres” or three-dimensional cultures that represent a micro-niche of the tumor believed to be enriched in stem-like cells [28]. While we have previously shown that anti-EMP2 treatment can reduce the tumor initiation frequency [14], its effects on taxane-resistant cells remain unexplored. Treatment with anti-EMP2 mAbs numerically reduced the number of spheres in culture compared to the control (Figure 5C), further suggesting that anti-EMP2 treatment can reduce this stem-like niche.

To extend these results using in vivo models, we combined anti-EMP2 mAbs with taxane-based chemotherapy in an orthotopic MDA-MB-231 model. Mice were treated twice weekly with 10 mg/kg anti-EMP2 or control mAbs and weekly with 20 mg/kg docetaxel or a combination of anti-EMP2 mAbs and docetaxel. Both the bi-weekly anti-EMP2 therapy and weekly docetaxel administration reduced tumor load (Figure 5D). While mice were treated with docetaxel and the antibodies on alternate days, docetaxel treatment alone did produce significant toxicity. However, in contrast to single-agent administration, combination therapy produced a sustained, significant response, aligning with the idea that EMP2 may serve as a molecular target for taxane-resistant TNBC (Figure 5D, right; *p* = 0.008, Log Rank Mantel–Cox Test). Additional experiments will be needed to determine the dosing and timing for this combination therapy to perhaps reduce the toxicity associated with taxane treatment.

## 4. Discussion

In this study, we complete a comprehensive analysis on the potential of EMP2 as a functional biomarker for taxane resistance in TNBC. We observed an increase in the number of strongly EMP2-positive cells in both animal models and patient samples of TNBC following taxane-based chemotherapy, with higher EMP2 expression correlating to diminished overall and progression-free survival. This increased expression suggested that EMP2 may not only serve as a biomarker for monitoring TNBC progression but also as a potential therapeutic target. To next determine if anti-EMP2 mAbs could serve as a treatment option for taxane-resistant disease, we treated tumors with anti-EMP2 mAbs sequentially or concurrently in combination with docetaxel, demonstrating promising results in both syngeneic and xenograft models.

Our results strongly suggest a correlation between docetaxel treatment and EMP2 tumor expression, whereby docetaxel increases EMP2 levels. Of note, EMP2 expression, as evaluated by immunohistochemical staining, is largely restricted to the tumor parenchyma, whereas the tumor stroma (endothelial cells, immune cells) show below-detection levels of expression. However, while we can show an increase in the number of strongly EMP2-positive tumor cells, we cannot show causation between treatment and antigen expression. While it is possible that taxanes increase the expression of the EMP2 transcript or protein levels, it is also possible that the increase in strongly EMP2-positive cells occurs through a selection process. As EMP2 expression can be heterogeneous within the naïve tumor parenchyma, it is possible that strongly EMP2-positive cells are more resistant to taxane treatment, thus leaving the tumor mostly populated with “high-EMP2” cells after taxane treatment. Additional experiments will be needed to evaluate both possibilities.

Taxanes, unlike many other anti-neoplastic drugs which damage RNA or DNA, promote cell death by inhibiting microtubular depolymerization. This action results in the detention of cells in the G2/M mitotic phase [22]. The success of taxanes in reducing tumor burden is evident given their approval for the treatment of breast, lung, gastric, and ovarian cancers [4]. Nonetheless, resistance remains a challenge. While taxane resistance is likely multifactorial, the upregulation of the transcription factor HIF-1a, a critical regulator of the hypoxic response, has been shown to protect BC cells from taxane treatment-induced apoptosis [29]. In response to hypoxia, HIF-1a stabilizes and translocates to the nucleus, initiating the transcription of multiple genes, including those involved in autophagy, glucose metabolism, and mitochondrial respiration [30]. Mechanistic studies have suggested that taxanes generally, and HIF-1a activation specifically, induce a transition to a CSC-like phenotype defined by CD44^Hi^ and variable levels of CD24 expression; this phenotype correlates with activation of the Src family kinase (SFK)/Hck pathways [31,32]. While additional experiments will be needed to decipher the mechanism by which taxanes specifically increase EMP2, the activation of the SFK and HIF-1a pathways in taxane-resistant cells is intriguing, as our previous results have shown that EMP2 is upstream of these pathways [15,33]. Several reports by our group and others suggest that EMP2 alters cellular adhesion and the subsequent modulation of FAK and Src activation, directly altering cellular phenotypes, such as invasion under normoxic conditions [33,34]. Under hypoxic conditions, EMP2 promotes the expression of HIF1a, driving cellular features associated with the CSC phenotype and neoangiogenesis [14,33].

TNBC is a heterogenous disease, demonstrating diversity in histologic patterns and subtypes, with transcriptional profiling suggesting that there are six distinct TNBC subtypes, including immunomodulatory (IM), mesenchymal (M), mesenchymal stem-like (MSL), luminal AR (LAR), and two basal-like (BL1 and BL2) subtypes [35]. While a significant proportion of TNBC tumors are highly sensitive to chemotherapy [36], subtypes demonstrate differential sensitivity to chemotherapy and targeted agents. For example, BL1 cell lines show sensitivity to genotoxic agents, whereas M cell lines respond to PI3K/mTOR treatment [35]. While it is still unknown if distinct subtypes respond more or less favorably to taxane treatment [37], several studies have shown that taxanes exhibit significant activity in all forms of TNBC and display both preclinical and clinical synergies when combined with other chemotherapeutics, such as platinum agents and Adriamycin [38]. However, while combination therapies can improve the pCR rate, this often comes at the cost of increased toxicity [2]. For TNBC patients who develop residual disease after chemotherapy, rates of metastatic recurrence remain high, leading to poor long-term prognosis [39]. 

Metastatic and recurrent phenotypes are responsible for 90% of deaths in patients with BC [40]. With current treatment protocols, the 5 year average survival rate in TNBC remains at only 65% in cases of node-positive disease and 11% for those with distant metastasis [41]. Thus, there is a need to find functional biomarkers to evaluate treatment response. Using the mammosphere assay as an in vitro surrogate for cells enriched for metastatic potential [42], we have previously shown that the loss of EMP2 expression is sufficient to reduce mammosphere formation [14]. Similarly, in this study, we show that treatment with anti-EMP2 mAbs significantly reduced the number of taxane-resistant mammospheres. In this manner, we speculate that targeting EMP2 may reduce the pool of viable cancer stem-like cells, thereby leading to a reduction in overall tumor burden.

## 5. Conclusions

Our results provide a functional rationale for the poor outcomes associated with high EMP2 expression in TNBC and open new avenues for targeted therapeutic strategies. The correlation between EMP2 upregulation, taxane resistance, and poor overall survival, coupled with the promising results of anti-EMP2 mAbs in reducing tumor burden, underscore the potential of targeting EMP2 to manage taxane-resistant TNBC. Further investigations are warranted to understand the exact mechanisms by which EMP2 may influence taxane resistance and the progression of TNBC. These studies should focus on the molecular interplay between EMP2, HIF-1a stabilization, and the activation of the Src family kinase pathways under normoxic and/or hypoxic conditions.

## Figures and Tables

**Figure 1 cancers-16-01481-f001:**
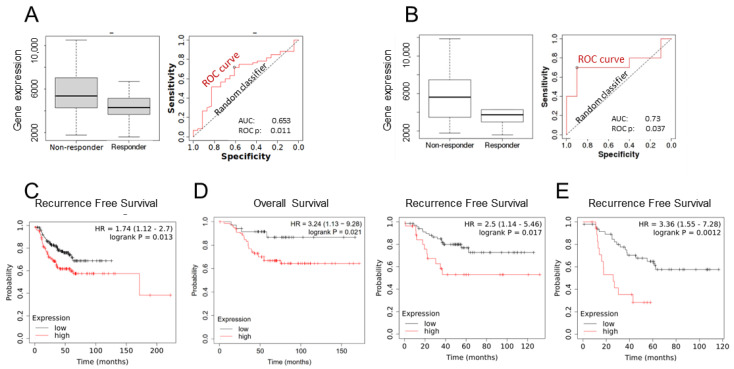
In silico analysis links high EMP2 transcript levels to taxane resistance and poor survival. (**A**) Following a standard chemotherapy protocol utilizing taxane treatment, 83 node-positive BCs showed a 1.2-fold increase in EMP2 expression (*p* = 0.03 by Mann–Whitney U test; ROC *p* = 0.011). Higher expression of EMP2 occurred in 72.3% (60/83) of patients who did not respond to this standard therapy compared to 27.7% (23/83) who did. (**B**) The data were further stratified to investigate EMP2 expression in node-positive TNBC. A total of 20 TNBC patient samples were stratified into two cohorts by high and low expression of EMP2 mRNA using ROC plotter [22]. Tumors from patients who failed to achieve a pathological complete response showed a 1.6-fold increase in EMP2 mRNA expression compared to tumors that responded to a taxane-based therapy (*p* = 0.09 by Mann–Whitney U test; ROC *p* = 0.037). (**C**) Using KM plotter, EMP2 was evaluated in 275 BC patients who received systemic chemotherapy. High expression of EMP2 correlated with poor relapse-free survival (*p* = 0.013). (**D**) Using KM plotter, EMP2 expression was stratified in 116 patients with TNBC (ER-/PR-/HER2-) who received any systemic chemotherapy. Left: high expression of EMP2 occurred in 69% of patients and correlated with poor overall survival (*p* = 0.021). Right: high expression of EMP2 correlated with poor relapse-free survival (*p* = 0.017). (**E**) When node-positive TNBC patients were assessed, high expression of EMP2 significantly correlated with poor relapse free survival (*p* = 0.001).

**Figure 2 cancers-16-01481-f002:**
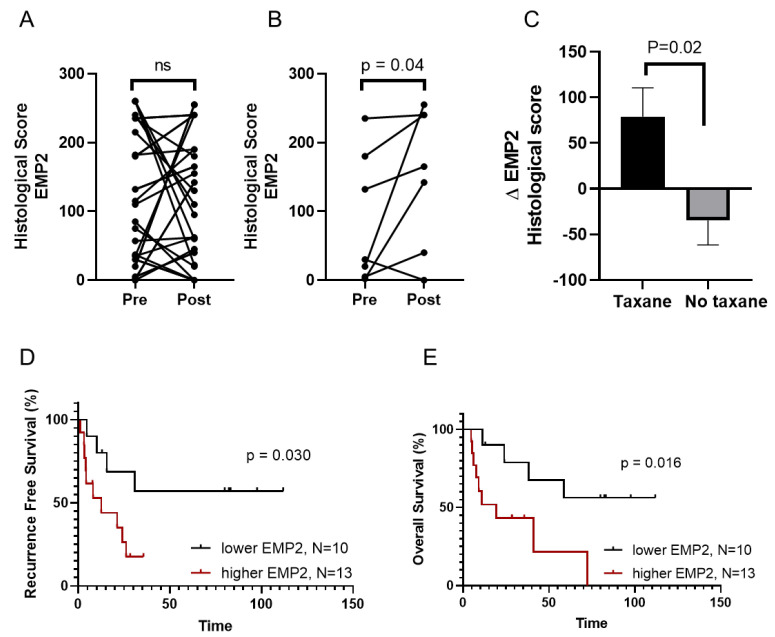
EMP2 protein expression is upregulated in taxane resistant tumors and correlates with reduced survival. (**A**) EMP2 protein levels were assessed in 23 matched pre and post treatment biopsies from a cohort of women with TNBC from Peru using standard immunohistochemistry. Left: no statistical significance (ns) was observed among the pre- and post-treatment patients, with a gain in EMP2 expression in 13/23 (56.5%) and a reduction in 10/23 (43.5%) patients (*p* = 0.89, Wilcoxon matched-pairs signed rank test). (**B**) Stratification of patients who received a taxane as part of their treatment. Post-taxane tumors showed higher expression of EMP2 than naïve tumors (*p* = 0.04, Wilcoxon matched-pairs signed rank test). N = 7. (**C**) Stratification of data based on treatment regimens. Patients who received a taxane showed significant upregulation of EMP2 compared to women who did not. (*p* = 0.02, Mann–Whitney U test). (**D**,**E**) The change in expression of EMP2 (higher delta vs. lower delta) was analyzed in all patients. N = 24. An increase in EMP2 following treatment correlated with reduced recurrence-free (**D**) and overall survival (**E**).

**Figure 3 cancers-16-01481-f003:**
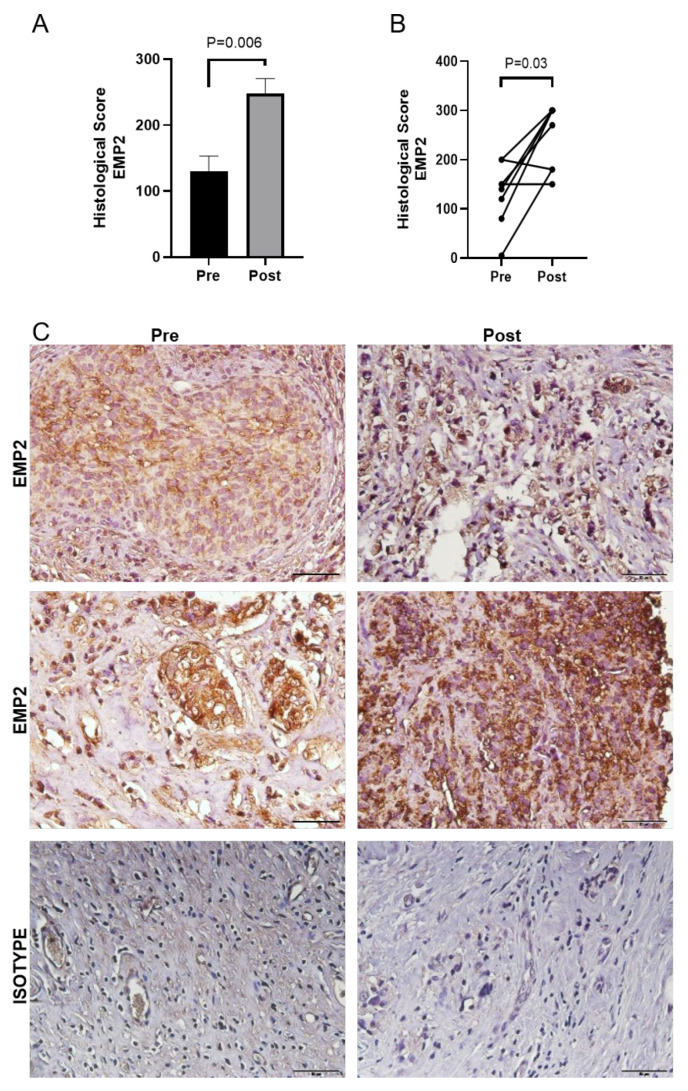
Validation of EMP2 expression in TNBC. (**A**) In an independent cohort of patients, EMP2 levels were correlated in naïve and post-neoadjuvant residual tumors. Patients post-chemotherapy showed on average higher expression of EMP2 compared to pre-treatment tumors. (**B**) Patient naïve and residual tumors were matched. The number of EMP2 strongly positive cells increased post neoadjuvant treatment in 6/8 patients (75%; *p* = 0.03, Wilcoxon matched-pairs signed rank test). (**C**) Representative EMP2 staining from two patients are shown, with an isotype control provided as comparison; N = 8 matched samples. Magnification: 400×. Scale bar = 50 μm.

**Figure 4 cancers-16-01481-f004:**
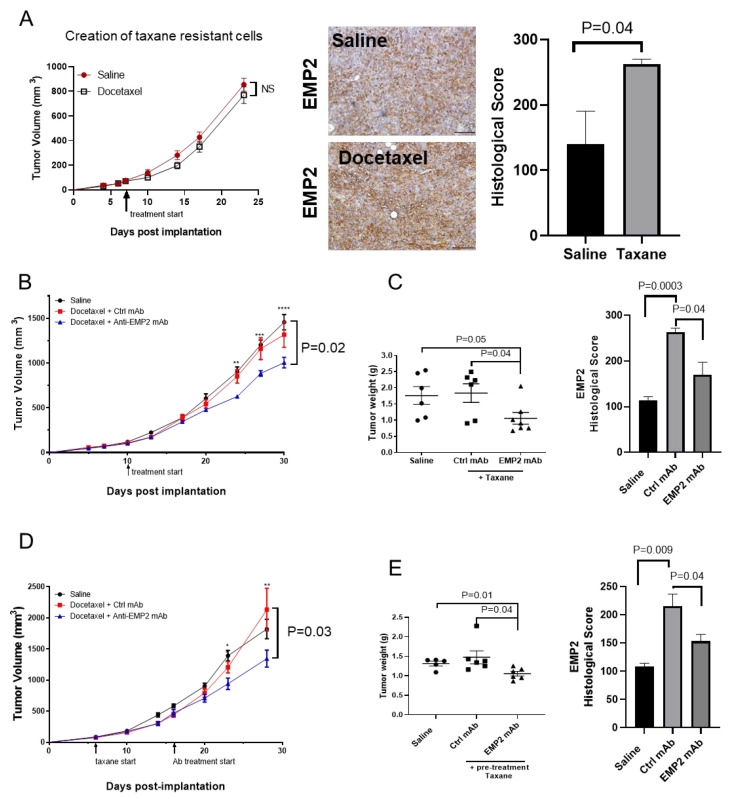
EMP2 serves as a functional biomarker for 4T1/FLuc tumors. (**A**) 4T1/FLuc tumors were established in the mammary fat pads of Balb/c mice and treated with saline or 10 mg/kg docetaxel weekly when tumors approached 100 mm^3^. N = 4. No significant differences were observed after three treatments (*p* = 0.25, Two-way ANOVA). Right: tumors were then analyzed using immunohistochemistry for EMP2 expression. The number of strongly EMP2-positive cells significantly increased following taxane treatment (N = 4, with the average ± SEM shown). Scale bar = 100 μm. NS = not significant. (**B**) Anti-EMP2 mAbs synergistically reduced tumor load in combination with docetaxel. Tumors were created as above and treated concurrently with 10 mg/kg docetaxel weekly and anti-EMP2 mAbs or control IgG at 10 mg/kg twice a week. A significant reduction in tumor load was observed (*p* = 0.02, Two-way ANOVA) N = 5. (**C**) At the conclusion of treatment, tumors were harvested and weighed. A significant reduction in tumor weight was observed in docetaxel plus anti-EMP2 mAbs compared to the other groups. N = 5. Right: EMP2 expression and localization were evaluated in tumors post-treatment. Taxane treatment increased the number of strongly EMP2-positive tumor cells while treatment with the anti-EMP2 mAb significantly reduced it. Groups were compared using Student’s *t* test. N = 3. (**D**) Sequential administration of anti-EMP2 or control mAbs to taxane-resistant tumors was tested. 4T1/FLuc tumors were initially treated with 10 mg/kg of docetaxel weekly. After 2 treatments, anti-EMP2 or control mAbs at 10 mg/kg was added to their treatment protocol twice a week. Compared to the control, anti-EMP2 mAbs in combination with docetaxel reduced tumor load and tumor weight (*p* = 0.03, Two-way ANOVA; N = 6 for treatment groups). (**E**) At the conclusion of treatment, tumors were harvested and weighed as above. A significant reduction in tumor weight was observed with the docetaxel plus anti-EMP2 mAb group compared to the others. N = 6 for treatment groups. N = 5 for the saline group. Right: EMP2 expression and localization was evaluated in tumors post treatment. Taxane treatment increased the number of strongly EMP2-positive tumor cells while treatment with the anti-EMP2 mAb significantly reduced EMP2 expression. Groups were compared using Student’s *t* test. N = 3. *, *p* ≤ 0.05; **, *p* ≤ 0.01; ***, *p* ≤ 0.001; **** *p* ≤ 0.0001.

**Figure 5 cancers-16-01481-f005:**
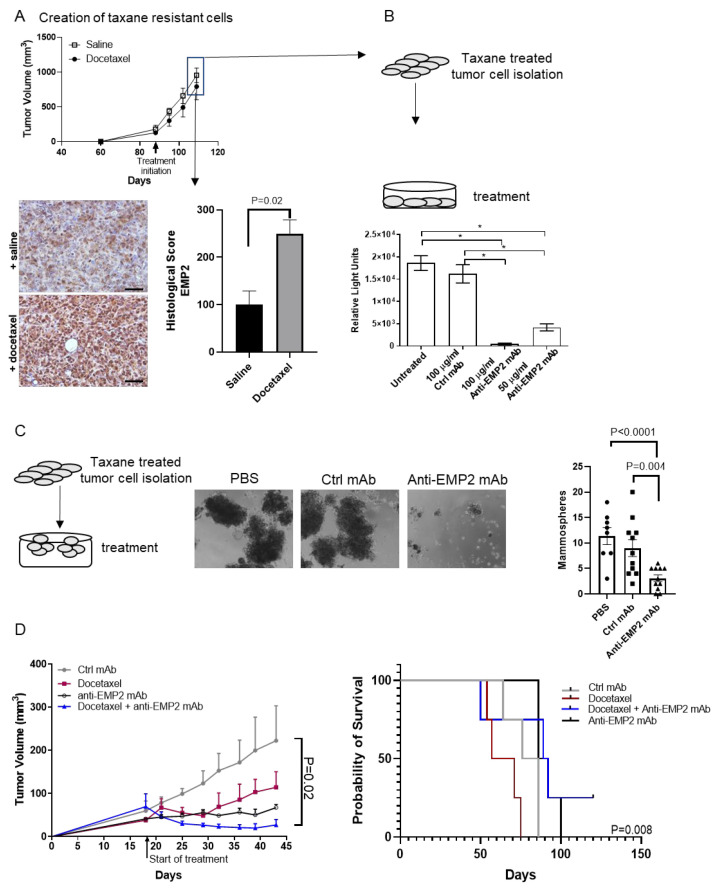
Anti-EMP2 mAbs treat taxane-resistant MDA-MB-231 tumors. (**A**) MDA-MB-231 cells were injected into the mammary fat pad of nu/nu mice and treated weekly with either docetaxel at 20 mg/kg or saline when tumors approached 200 mm^3^. After 110 days, tumors were harvested, processed for immunohistochemistry, and stained for EMP2. Docetaxel treatment significantly increased the number of EMP2-positive cells compared to control (*p* = 0.02). N = 3. Magnification = 400×. (**B**) Taxane-resistant tumors were harvested, and a single cell suspension was created. Cells were treated with either anti-EMP2 or control mAbs, and cellular viability was determined through the measurement of ATP. Treatment with 50 or 100 mg/mL of anti-EMP2 mAb significantly reduced viability. This experiment was repeated twice, with N = 3 per experiment. Results shown depict one representative experiment. *, *p* < 0.05. (**C**) To determine if anti-EMP2 mAbs altered mammosphere formation, cells collected above were grown under low-adhesion conditions. Compared to the control, a significant reduction in the number of mammospheres was observed. N = 8–11 wells from 3 mice. Results presented are the average ± SEM from all wells. Magnification = 100×. (**D**) MDA-MB-231 tumors were established in nu/nu mice and treated with docetaxel, control mAbs, anti-EMP2 mAbs, or a combination of anti-EMP2 mAbs with docetaxel when tumors approached 50–75 mm^3^. The largest reduction in tumor load was observed from the combination therapy. *p* = 0.03, Two-way ANOVA; N = 4. Right: the overall survival of animals treated in each group was evaluated. *p* = 0.008, Log-rank (Mantel–Cox) test.

**Table 1 cancers-16-01481-t001:** Clinical characteristics of pre- and post-NAC-treated TNBC.

EMP2 Increased after Chemotherapy (Yes/No)		*p*-Value
Age		
Median	46 years	0.47 §§
Range	29–69 years	
Stage		1 §
IIa	1	
IIb	2	
IIIb	19	
IIIc	1	
Lymph node spread		0.068 §
Present	16	
Absent	7	
Menopausal status		
Pre-menopausal	11	1 §
Post-menopausal	12	

§§ Spearman correlation; § Fisher exact test.

## Data Availability

The data that support the findings of this study are available on reasonable request from the corresponding author (M.W.).

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
