# Peer review of "EMP2 Serves as a Functional Biomarker for Chemotherapy-Resistant Triple-Negative Breast Cancer"

_cancers, 2024, doi:10.3390/cancers16081481_

Round 1

Reviewer 1 Report

Comments and Suggestions for Authors

The manuscript investigates the potential of EMP2 (Epithelial Membrane Protein 2) as a functional biomarker for taxane resistance in triple-negative breast cancer (TNBC). Through in-depth analyses using both animal models and patient samples, the study reveals elevated EMP2 levels following taxane-based chemotherapy, correlating with diminished overall survival in TNBC patients. The findings suggest EMP2 as a promising biomarker for monitoring TNBC progression and a potential therapeutic target. The authors generate anti-EMP2 monoclonal antibodies (mAbs) and they show promising results in reducing tumor burden in both syngeneic and xenograft models. The study highlights the correlation between EMP2 upregulation, taxane resistance, and poor outcomes in TNBC, emphasizing the potential of targeting EMP2 for managing taxane-resistant TNBC. To strengthen the manuscript, these following aspects could be considered:

While the study establishes correlations, it lacks in-depth mechanistic insights into how EMP2 influences taxane resistance at a molecular level.

the specificity and selectivity of anti-EMP2 monoclonal antibodies needs validation, any results show decreased EMP2 expression in EMP2-treated group?

Author Response

Thank you so much for your critical review of our paper.  We have attempted to address your concerns below and modified the manuscript accordingly.

The manuscript investigates the potential of EMP2 (Epithelial Membrane Protein 2) as a functional biomarker for taxane resistance in triple-negative breast cancer (TNBC). Through in-depth analyses using both animal models and patient samples, the study reveals elevated EMP2 levels following taxane-based chemotherapy, correlating with diminished overall survival in TNBC patients. The findings suggest EMP2 as a promising biomarker for monitoring TNBC progression and a potential therapeutic target. The authors generate anti-EMP2 monoclonal antibodies (mAbs) and they show promising results in reducing tumor burden in both syngeneic and xenograft models. The study highlights the correlation between EMP2 upregulation, taxane resistance, and poor outcomes in TNBC, emphasizing the potential of targeting EMP2 for managing taxane-resistant TNBC. To strengthen the manuscript, these following aspects could be considered:

While the study establishes correlations, it lacks in-depth mechanistic insights into how EMP2 influences taxane resistance at a molecular level.

We thank the reviewer for this important comment.  While this paper does not address how EMP2 influences taxane resistance at the molecular level, we have expanded the discussion to speculate on this idea.

the specificity and selectivity of anti-EMP2 monoclonal antibodies needs validation, any results show decreased EMP2 expression in EMP2-treated group?

We have provided images of EMP2 post-taxane treatment as well as additional references that show the specificity and selectivity of the anti-EMP2 monoclonal antibodies.  We have also included reference batch data on the antibody to show its purity.

Reviewer 2 Report

Comments and Suggestions for Authors

I am glad I had the opportunity to review this article which shows great promise in using EMP2 as diagnostic marker and therapeutic target for TNBC. However, the human data analysis on publicly available data and retrospective cohorts needs clarification as it doesn’t seem to be focussing on TNBC only.  The results of the in vitro/in vivo study are very interesting but could do with some improvement (e.g. missing control). The discussion could also be improved to better highlight the results.

I would recommend the following changes:

-          - The list of references isn’t in the same order as in the main text of the article.

 - Introduction: I would suggest introducing the various subtypes of TNBC as they might explain the different response to chemotherapy. It could also be discussed at the end of the manuscript.

- Please specify if EMP2 is expressed or not in normal breast tissue (fig S1 shows that it is weakly if not at all expressed in normal breast tissue, please refer to it).

-        -  Line 109: please add more details on how the scoring was performed. Is EMP2 expressed in both tumor and stromal cells? If yes, did it affect the scoring?

-         - Results 3.1: this part is a bit confusing as this work focusses on TNBC and yet, other subtypes were included in the analysis (fig 1A, HER2- which can include luminal A and B) and presence of non-TNBC in the Peruvian cohort (table 1). Moreover, it is not clear how many TNBC were present in the Peruvian cohort (table 1) as not all basal breast cancers are TNBC. Could the authors re-do the analysis including only TNBC

-     -  Fig 1: the details provided in the supplementary data are not enough to reproduce fig 1 graphs using publicly available data. Please provide a more detailed description of the criteria chosen to obtain the ROC and KM plots (e.g screenshot of the ROC/KM plotter first page).

-         -  Line 190: p=0.0068 but in table 1, p=0.068.

 - Table 1: the authors rightly mentioned in the introduction that TNBC strongly affect Black and Hispanic women. Seeing that table 1 refers to a Peruvian cohort, I would suggest adding the racial/ethnicity details, unless it is showed in a previous publication, please, refer to it after re-organising your list of references.

 - Table S1 is missing.

-         -  Fig 3 C: the pictures are too small to see details such as stromal or tumor cells with low expression of EMP2. Please consider adding a supplementary figure with larger pictures.

-          - Figure 2A and 4 and 5A: please increase the font size of the graphs to make these very interesting data easier to read.

-         -  Line 242-244: although I agree that more tumor cells express high levels of EMP2 after taxane treatment, the results do not prove that taxane directly increases the expression of EMP2 protein in TNBC cells.

The results could be explained by a selection process: since there are cells with high expression of EMP2 in the saline control, it is possible that high expression of EMP2 confers resistance to taxane in TNBC cells. Cells with lower expression of EMP2 would be killed by taxane, leaving tumors mostly populated with “high-EMP2” cells after taxane treatment. Please comment on that in the manuscript.

-         -  Please change line 244: the data don’t show an increase in protein expression but an increase in cells with high EMP2 expression.

-        -   line 277: “As shown in figure 5A, MDA-MB-231 tumors were generally insensitive to taxane treatment”. Fig 5A doesn’t show tumour growth curve for the saline control, could you please add it.

The immunostaining pictures are a bit small. 

-         -  Line 277-278: please remove “replicating findings…chemotherapy”. One cell line doesn’t represent 30% of TNBC. One would have to test several cell lines to affirm that their resistance to taxane replicates patient’s data.

-         - Line 296-298: please add the reference of the paper showing the statement.

-    - Figure 5D: for clarity, please give more details, on the graph or in the supplementary data, on which days docetaxel and anti-EMP2 mAB were injected in the combined treatment (was docetaxel injected the same day as one of the weekly injection of anti-EMP2 mAB?).

-         - Figure D survival graph: the data on this graph are very interesting and could do with a bit more discussion.

- Please comment on the effect of docetaxel alone on mouse survival. Is it possible that the lower survival in docetaxel-treated group is due to general toxicity of the drug at 20mg/kg?

Survival is much better in the combined treatment, suggesting that the anti-EMP2 mABs counteract docetaxel toxicity. Could you please comment on that in your manuscript.

-         - Discussion: the results, especially the in vivo/in vitro ones, could be discussed in more depth to highlight their importance in developing EMP2 as a prognostic factor and a targeted therapy for TNBC

-          - Line 377: same comment as for line 244, please change “EMP2 upregulation” for cells with high expression of EMP2 or something similar.

-        -   Fig S1: Figure S5 in the legend? I think there is a typo for the scale bar, is it 100µm instead of 100mm?

-         -  Figure S2: not mentioned in the text.

    If the tumour cells were injected into mice as supplementary material and methods mention, please change “4T1 tumours were implanted” for “4T1 cells were injected”.

There seem to be a typo at the end of the legend: isn’t it Figure 4C?

Round 2

Reviewer 2 Report

Comments and Suggestions for Authors

Many thanks for making the changes suggested in the first report, they have greatly improved the reading of your manuscript and highlighted the importance of the data.

I only have minor comments about the supplementary figures:

- line 251-255: is it fig S1?

- line 346: is it fig S2A?

- fig S3 and S4 are missing

Please add these figures and double-check they are mentioned appropriately in the text.
